# Status of psychological health of students following the extended university closure in Bangladesh: Results from a web-based cross-sectional study

Md. Jamal Hossain[1]*, Foyez Ahmmed[2]*, Labony Khandokar[3], S. M. Abdur Rahman[4], Asaduzzaman Hridoy[2], Farhana Alam Ripa[5], Talha Bin Emran[6], Md. Rabiul Islam[7], Saikat Mitra[8], Morshed Alam[9]

1 Department of Pharmacy, State University of Bangladesh, Dhanmondi, Dhaka, Bangladesh, 2 Department of Statistics, Comilla University, Kotbari, Cumilla, Bangladesh, 3 Department of Pharmacy, East West University, Dhaka, Bangladesh, 4 Department of Clinical Pharmacy and Pharmacology, Faculty of Pharmacy, University of Dhaka, Dhaka, Bangladesh, 5 Department of Pharmacy, Brac University, Mohakhali, Dhaka, Bangladesh, 6 Department of Pharmacy, BGC Trust University Bangladesh, Chittagong, Bangladesh, 7 Department of Pharmacy, University of Asia Pacific, Farmgate, Dhaka, Bangladesh, 8 Department Pharmacy, Faculty of Pharmacy, University of Dhaka, Dhaka, Bangladesh, 9 Institute of Education and Research, Jagannath University, Dhaka, Bangladesh

* jamal.du.p48@gmail.com, jamalhossain@sub.edu.bd (MJH); foyez.sbi@gmail.com (FA)

**Data Availability Statement:** The total 16 items in PHQ-9 and GAD-7 scales are shown in the

## Abstract

Students' severe affective mental distress has emerged as significant public health attention globally because of the disastrous effects of coronavirus disease 2019 (COVID-19). The current study aimed at exploring the prevalence of two alarming psychological distresses, depression and anxiety, among university students following a prolonged shutdown of educational institutions in Bangladesh. A cross-sectional online-based study was conducted by deploying two standard scales to assess the depression and anxiety among Bangladeshi students from various universities amid the 2nd stream of the COVID-19 pandemic. A total of 568 Bangladeshi university students participated in this questionnaire-based survey through various social media platforms. Frequency and percentage distribution as univariate, chi-square ($\chi^2$) test as bivariate, and logistic regression as multivariate analyses were applied to investigate the prevalence of depression and anxiety and their associated various sociodemographic factors. After cleaning and eliminating the partial data, we analyzed 465 responses, where 42% were female and 64.3% were from public universities. Both mental disorders were prevalent in more than 50% of Bangladeshi university students. The students from the private universities were two times and 2.7 times more depressed and anxious, respectively than the students from the public universities. In addition, the students who became incomeless had significantly more anxiety (adjusted odds ratio [AOR] = 1.711; $p$ = 0.018) than those who did not lose income source during the COVID-19 lockdown. The present study revealed that more than 50% of Bangladeshi university students were suffering from depression and anxiety. Several effective measures must be assured by the concerted efforts of university authorities, educationalists, and the Government to alleviate these distressing mental health burdens.

supplementary S1 Table. Besides, all the raw or extracted datasets in this research are available in the manuscript and supplementary files.

**Funding:** The author(s) received no specific funding for this work.

**Competing interests:** The authors have declared that no competing interests exist.

## Introduction

The highly transmissible pathogenic agent, severe acute respiratory syndrome coronavirus 2 (SARS-CoV-2), originated from Wuhan, China, caused a disastrous outbreak of coronavirus disease 2019 (COVID-19) that has already jeopardized the entire world [1–3]. The World Health Organization addressed this unforeseen public health emergency, a pandemic on March 11, 2020 [1, 2, 4]. Following the first identification of COVID-19 patients on March 8, 2020, Bangladesh declared to shut down public mobility and activities on March 26, 2020, including educational institutions (educational institutions have been closed from March 18, 2020), to impede the rapid community transmission of the virus [4]. Lockdown has been the preventive measure to attenuate the propagation of this novel nosogenic disease that causes both physical and mental uneasiness [5, 6]. More extended restrictions to social gatherings, quarantine, and isolation procedures effectuated a tormenting situation, including anxiety, loneliness, and depression among the population, especially students [7]. Furthermore, COVID-19 lockdown increased the total number of social media users as well as their extent of use, and such extended use of modern devices may attribute to changes in a regular routine, sleep patterns, and mental conditions [8]. Several studies reported that the lockdown and associated restrictions (e.g., online schooling, social distancing) imposed due to the COVID-19 pandemic has resulted in mental health impairments [9–11]. Young people are more likely to be more sufferers of psychological disorders than adults, and they are more vulnerable to the adverse effects of social isolation, including educational institution closure [11]. A meta-analysis on 29 studies (80 979 youth) showed that the global prevalence of clinically increased child and adolescent depression and anxiety was 25.2% and 20.5%, respectively [9]. Besides, Humer et al. [10] reported that school reopening and reduced social distancing measures were associated with improved mental health conditions among high school students.

The current pandemic outbreak has caused modification of psychological demeanor through the expression of tension, anxiety, and depression [12, 13]. Moreover, symptoms of normal flu and other infectious disorders, like fever, cough, hypoxia, and insomnia, can augment such anxiety and apprehension levels [14]. A recent systematic review claimed that the manifestation of stress, anxiety, and depression was 29.6%, 31.9%, 33.7%, respectively, among the Asian and European populations [15]. Furthermore, a cross-sectional survey reported that young adults aged less than 35 years pursued daily news of COVID-19 for greater than 3 hours and demonstrated uprising levels of stress and anxiety compared to the adults above 35 of age who were less interested in COVID-19-related news [16].

The education sector is the second most affected sector after the economic sector due to the COVID-19 outbreak [17]. On account of residing longer time at home, lingering work hours, participation in homeschooling, and exacerbated health risks, many individuals' daily life and sleeping patterns have been invaded deleteriously [16]. Like other developing countries, Bangladesh has been fighting extreme socio-emotional challenges, resulting in the stimulation of several psychological distresses, including depression, anxiety, stress, sleep disorders, and subsequently augmenting fear and suicidal behaviors [7, 18]. However, compared to other people, university students are enduring excessive pressure and being susceptible to severe mental illness on account of the closure of educational institutions [18].

Moreover, low and middle-income people who earn quotidian wages lost their works and suffered from a severe economic crisis [19]. Amidst this extreme difficulty, many students, could not bear their huge academic expenses. Many students were compelled to drop one or two semesters due to the high costs and apprehension about the digital education system. Many students completely dropped their academic life in rural areas and tried to earn to support their families. Many of the female students had to face child or early marriage during this

pandemic situation [20]. In urban areas, most university students who had earned money through private tuition to bear their personal and educational expenses lost their source of income [21]. Consequently, several mental health complications have arisen extremely among Bangladeshi university students. It has recently been reported that around 50%-60% of Bangladeshi university students were suffering from underprivileged online education, severe session jam phobia, and extreme psychological distress [18]. In a systematic review, Mamun et al. [22] stated that 46.92% to 82.4%, 26.6% to 96.82%, and 28.5% to 70.1% of Bangladeshi students faced mild to severe symptoms of depression, anxiety, and stress, respectively during COVID-19 lockdown. Several socioeconomic factors associated with negative emotions exacerbated by the extended COVID-19 lockdown have intensely impacted the mental health conditions of all ages, particularly students, due to the university closure for a prolonged time [18, 23]. Ela et al. [21] reported that the graduation delay had been mainly occurred due to the lengthy university closure that has significantly increased various mental health complications, including depression and anxiety, among the tertiary level of students in Bangladesh.

Moreover, Bangladesh was suffering from the 2nd stream of COVID-19 outbreak from the last months of 2020 that has worsened the students' mental complications [24]. Therefore, in this paper, we aim to investigate the influences of various socio-demographic factors potentially responsible for enhancing several psychological problems in terms of depression and anxiety endured by university students in Bangladesh due to the extended COVID-19 shutdown.

## Methods

### Study design and sampling

A cross-sectional Google Form survey through social media platforms was conducted by designing a total of 32 questions to explore the two most frequent mental disorders, depression and anxiety, among Bangladeshi university students (bachelor and master's) during the COVID-19 lockdown in Bangladesh. The questionnaire was segmented into three sections where section A consisted of the sociodemographic information of the participants, and sections B and C were designed with depression and anxiety-related questions [7]. After the initial drafting of the questionnaire in English, it was then translated into in Bangla version. Then the questionnaire was checked and validated by a bilingual medical expert through a forward and backward translation process. A simple snowball sampling strategy [7, 25] was adopted for collecting nationwide data rapidly from November 20 to 30 of 2020. The survey was introduced, having some information about the research topic, its importance, and ethical considerations. The students who were never clinically diagnosed with mental distortion before COVID-19 were included in the study. Respondents could use their smartphone, personal computer, or tablet to complete the survey voluntarily. Informed consent was sought virtually from each of the respondents before participation in the survey. All the research guidelines were set and obeyed according to the regulations of the World Medical Declaration of Helsinki (WMA, 2018). Besides, the Ethics Committee of the FBS (Faculty of Biological Science), University of Dhaka, Bangladesh, critically reviewed and formally approved the study protocols and procedures (Ref. No. 120/Biol. Scs.).

### Data collection and analysis

Within the short data collection period, a total of 568 participants participated in this survey, and we carefully preserved all the collected data confidential and private. After cleaning and eliminating the partial data, we analyzed 465 responses, where the statistical significance level was 5%, i.e., $p < 0.05$, for all the statistical methods, including chi-square ($\chi^2$) test and logistic

regression analysis [26]. For univariate analysis, we calculated descriptive statistics (frequency and percentage distributions) of the background characteristics of the participants. For finding the association of different factors with depressive and anxiety disorders, we used the chi-square test of association. Finally, a binary logistic regression was used to find the potential factors associated with depression and anxiety disorder after adjusting other factors. All the analyses have been conducted using the software IBM SPSS (version 20).

### Covariates

We considered gender (male, female), type of university (public, private, and others), study year (1st year, 2nd year, 3rd year, 4th/5th year, masters), change in family income (less than before, same as before, and more than before), spending lockdown period with family (yes, no), living area during lockdown (urban, rural), occupation of the family head (government job, private job, own business, farming, and others), lost your way of income like tuition (yes, no), Afraid of completing graduation in time (yes, no) as covariates of the study.

### Assessment of depression and anxiety

The effects of prolonged COVID-19 lockdown on psychological health in terms of depression and anxiety of the current university students of Bangladesh was screened utilizing the two validated scales, PHQ-9 (PHQ = Patient Health Questioner) and GAD-7 (GAD = Generalized Anxiety Disorder) scales, respectively [7, 27]. The total 16 questions used in both scales (PHQ-9 and GAD-7) were stated in S1 Table. Items in both scales were designed as '0 = not at all', '1 = several days', '2 = more than half of the days', and '3 = nearly every day'. The participants, who attained cut-off scores $\geq$ 10 out of 27 on the PHQ-9 scale (Min to the max: 0 to 27) and $\geq$ 8 out of 21 on the GAD-7 scale (Min to the max: 0 to 21), were defined as having depression and anxiety, respectively [28]. Besides, we enumerated Cronbach's alpha reliability coefficient 0.774 and 0.817 for PHQ-9 and GAD-7, respectively, which indicated an excellent internal consistency of both scales [29].

## Results

### Demographic characteristics

The total questionnaire was segmented into four sections: demographic, PHQ-9, GAD-7, and several vital and concerning parameters for assessing the psychological distress conditions of the university students following the one year of COVID-19 shutdown in Bangladesh. Among the total 568 participants, 465 students (mean age 22 years, interquartile range, IQR = 20–23) responded completely in the survey where 58.1% (n = 270) were male, and 29.5% (n = 137) participated from various private educational institutions (Table 1). Around third-fourth of the respondents (n = 316; 68%) were from the 20–25 age group, and 31.8% (n = 148) of students were reading in higher grade class (4th/5th or higher) in the universities. Near 70% of the respondents responded from outside of Dhaka (capital city of Bangladesh), and around one-third of the students' family heads were government job holders. Similarly, all other demographic variables with their descriptive statistics were tabulated in Table 1.

### Chi-square ($\chi^2$) analysis of depression and anxiety

The current study reports above 50% of the students had depression (n = 254; 54.6%) and anxiety (n = 249; 53.5%) during the rapid rising period of the second stream of the current pandemic in Bangladesh. The $\chi^2$ test revealed that the several demographic variables 'change of family income,' 'type of the educational institutions,' and 'passing institutional closure time

**Table 1. Bangladeshi university students' sociodemographic characteristics and covariates with descriptive statistics (N = 465).**

| Parameters/Questions | Options | N | % |
|---|---|---|---|
| **Your gender =?** | Male | 270 | 58.1 |
| | Female | 195 | 41.9 |
| **Your age (years) =?** | < 20 | 127 | 27.3 |
| **(Mean = 22; IQR = 20–23)** | 20 to 25 | 316 | 68.0 |
| | > 25 | 22 | 4.7 |
| **Your university type =?** | Public university | 299 | 64.3 |
| | Private university | 137 | 29.5 |
| | Others | 29 | 6.2 |
| **Your study year =?** | 1st year | 98 | 21.1 |
| | 2nd year | 131 | 28.2 |
| | 3rd year | 88 | 18.9 |
| | 4th/5th year | 80 | 17.2 |
| | Master's | 68 | 14.6 |
| **Your current location =?** | Dhaka | 143 | 30.8 |
| | Outside of Dhaka | 322 | 69.2 |
| **Your living area during lockdown =?** | Rural | 227 | 48.8 |
| | Urban | 238 | 51.2 |
| **Your total family members =?** | ≤ 4 | 184 | 39.6 |
| **(Mean = 5.27; IQR = 4–6)** | ≥ 5 | 281 | 60.4 |
| **Afraid of completing graduation in time** | Yes | 385 | 82.8 |
| | No | 80 | 17.2 |
| **Anxious about getting a job in the future** | Yes | 404 | 86.9 |
| | No | 61 | 13.1 |
| **Spending lockdown period with family** | Yes | 133 | 28.6 |
| | No | 332 | 71.4 |
| **Occupation of family head** | Government job | 158 | 34.0 |
| | Private job | 106 | 22.8 |
| | Own business | 124 | 26.7 |
| | Farming | 39 | 8.4 |
| | Other | 38 | 8.2 |
| **Situation of family income** | More than before | 19 | 4.1 |
| | Less than before | 329 | 70.8 |
| | Same as before | 117 | 25.2 |
| **Has any family member lost job due to the COVID-19 outbreak** | Yes | 116 | 24.9 |
| | No | 349 | 75.1 |
| **Have you lost your way of income like tuition or any other job** | Yes | 318 | 68.4 |
| | No | 147 | 31.6 |

**Note:** N = Number of participants, % = percentage, IQR = Interquartile Range, MDD = Major Depressive Disorder, PHQ = Patient Health Questioner, GAD = Generalized Anxiety Disorder. PHQ (1 to 9) and GAD (1 to 7) questions were asked, based on "over the last two weeks, how many times the participants bothered by the stated problems".

with family' were significantly associated with both mental disorders ($p < 0.05$) (Table 2). The male students numerically suffered more from MDD (n = 155 (57.4%) vs. n = 99 (50.8%); $p = 0.156$) and GAD (n = 162 (60.0%) vs. n = 87 (44.6%); $p = 0.002$) compared to the female students. It is vital to note that the private university students were 15.8% more depressed (65% vs. 49.2%) and 18.2% more anxious (65.7% vs. 47.5%) than public university students. The

**Table 2.** $\chi^2$ analysis for assessing the association of different sociodemographic factors with depression and anxiety among Bangladeshi university students during the COVID-19 lockdown.

| Variables | Categories | Depression | | | | | Anxiety | | | | |
|---|---|---|---|---|---|---|---|---|---|---|---|
| | | No = 211 (45.4%) | | Yes = 254 (54.6%) | | | No = 216 (46.5%) | | Yes = 249 (53.5%) | | |
| | | N | % | N | % | p-value | N | % | N | % | p-value |
| **Gender** | Male | 115 | 42.6 | 155 | 57.4 | 0.156 | 108 | 40.0 | 162 | 60.0 | **0.001** |
| | Female | 96 | 49.2 | 99 | 50.8 | | 108 | 55.4 | 87 | 44.6 | |
| **Age (years)** | < 20 | 62 | 48.8 | 65 | 51.2 | 0.558 | 65 | 51.2 | 62 | 48.8 | 0.426 |
| **(mean = 22; IQR = 20–23)** | 20 to 25 | 138 | 43.7 | 178 | 56.3 | | 142 | 44.9 | 174 | 55.1 | |
| | > 25 | 11 | 50.0 | 11 | 50.0 | | 9 | 40.9 | 13 | 59.1 | |
| **University type** | Public | 152 | 50.8 | 147 | 49.2 | **0.006** | 157 | 52.5 | 142 | 47.5 | **0.002** |
| | Private | 48 | 35.0 | 89 | 65.0 | | 47 | 34.3 | 90 | 65.7 | |
| | Others | 11 | 37.9 | 18 | 62.1 | | 12 | 41.4 | 17 | 58.6 | |
| **Study Year** | 1st year | 44 | 44.9 | 54 | 55.1 | 0.334 | 47 | 48.0 | 51 | 52.0 | 0.907 |
| | 2nd year | 59 | 45.0 | 72 | 55.0 | | 61 | 46.6 | 70 | 53.4 | |
| | 3rd year | 38 | 43.2 | 50 | 56.8 | | 41 | 46.6 | 47 | 53.4 | |
| | 4th/5th year | 44 | 55.0 | 36 | 45.0 | | 39 | 48.8 | 41 | 51.2 | |
| | Master's | 26 | 38.2 | 42 | 61.8 | | 28 | 41.2 | 40 | 58.8 | |
| **Current location** | Dhaka | 63 | 44.1 | 80 | 55.9 | 0.703 | 68 | 47.6 | 75 | 52.4 | 0.751 |
| | Outside of Dhaka | 148 | 46 | 174 | 54 | | 148 | 46 | 174 | 54 | |
| **Living area during lockdown** | Rural | 92 | 40.5 | 135 | 59.5 | **0.040** | 97 | 42.7 | 130 | 57.3 | 0.116 |
| | Urban | 119 | 50.0 | 119 | 50.0 | | 119 | 50.0 | 119 | 50.0 | |
| **Total family members** | ≤ 4 | 86 | 46.7 | 98 | 53.3 | 0.633 | 96 | 52.2 | 88 | 47.8 | **0.045** |
| | ≥ 5 | 125 | 44.5 | 156 | 55.5 | | 120 | 42.7 | 161 | 57.3 | |
| **Spending lockdown period with family** | No | 50 | 37.6 | 83 | 62.4 | **0.033** | 47 | 35.3 | 86 | 64.7 | **0.002** |
| | Yes | 161 | 48.5 | 171 | 51.5 | | 169 | 50.9 | 163 | 49.1 | |
| **Occupation of family head** | Government Job | 74 | 46.8 | 84 | 53.2 | 0.763 | 81 | 51.3 | 77 | 48.7 | 0.221 |
| | Private Job | 48 | 45.3 | 58 | 54.7 | | 50 | 47.2 | 56 | 52.8 | |
| | Own Business | 55 | 44.4 | 69 | 55.6 | | 47 | 37.9 | 77 | 62.1 | |
| | Farming | 20 | 51.3 | 19 | 48.7 | | 18 | 46.2 | 21 | 53.8 | |
| | Others | 14 | 36.8 | 24 | 63.2 | | 20 | 52.6 | 18 | 47.4 | |
| **Change in family income** | More than before | 10 | 52.6 | 9 | 47.4 | **0.042** | 8 | 42.1 | 11 | 57.9 | **0.025** |
| | Less than before | 137 | 41.6 | 192 | 58.4 | | 141 | 42.9 | 188 | 57.1 | |
| | Same as before | 64 | 54.7 | 53 | 45.3 | | 67 | 57.3 | 50 | 42.7 | |
| **Lost the way of income** | Yes | 146 | 45.9 | 172 | 54.1 | 0.733 | 134 | 42.1 | 184 | 57.9 | **0.006** |
| | No | 65 | 44.2 | 82 | 55.8 | | 82 | 55.8 | 65 | 44.2 | |
| **Afraid of completing graduation in time** | Yes | 172 | 44.7 | 213 | 55.3 | 0.505 | 176 | 45.7 | 209 | 54.3 | 0.484 |
| | No | 39 | 48.8 | 41 | 51.2 | | 40 | 50.0 | 40 | 50.0 | |
| **Anxious about getting a job in the future** | Yes | 183 | 45.3 | 221 | 54.7 | 0.930 | 186 | 46 | 218 | 54 | 0.647 |
| | No | 28 | 45.9 | 33 | 54.1 | | 30 | 49.2 | 31 | 50.8 | |

rural areas' students were significantly more depressed than the urban areas' students during the lockdown period (n = 135 (59.5%) vs. n = 119 (50%); $p$ = 0.040). Besides, the students living outside of family were significantly more depressed (62.4% vs. 51.5%; $p$ = 0.033) and anxious (64.7% vs. n = 49.1%; $p$ = 0.002) than the students who were spending with families during the lockdown. Moreover, the income-losing students during lockdown were significantly more anxious than the students having the source of earning during the lockdown period (57.9% vs. 44.2%; $p$ = 0.006). Finally, all other factors potentially correlated with depression and anxiety with their corresponding bivariate analyses were listed in Table 2.

## Logistic regression analysis of depression and anxiety

During multivariate analysis, binary logistic regression was applied for exploring the significantly influencing variables after adjusting other parameters that may accelerate the psychological distress conditions of university students in Bangladesh. The private university students were suffering from two times (adjusted odds ratio [AOR] = 2.009, 95% Confidence Interval [CI] = 1.207 to 3.344; $p$ = 0.007) and 2.7 times (AOR = 2.699, 95% CI = 1.586 to 4.593; $p$ < 0.0001) depression and anxiety, respectively than the respondents from public institutions (Figs 1 and 2). It was found that the male students had more anxiety than female students (AOR = 1.551, 95% CI: 1.022, 2.353; $p$ = 0.039), and master's students were more likely to be depressed compared to bachelor students (for example, AOR of 4th/5th year students = 0.034, 95% CI: 0.165, 0.722; $p$ = 0.005). On the other hand, the students with $\geq$ 5 family members were significantly more anxious than those with $\leq$ 4 family members (AOR = 0.657, 95% CI: 0.437, 0.989; $p$ = 0.044). It can be highlighted that the students passing lockdown period due to COVID-19 apart from families had a significantly higher risk to have depression (AOR = 1.577, 95% CI: 1.019, 2.439; $p$ = 0.041) and anxiety (AOR = 1.89, 95% CI: 1.207, 2.967; $p$ = 0.005) than the students who were living with families. Besides, the respondents from the income-losing group had significantly more anxiety compared to the participants who had income source during the COVID-19 lockdown (AOR = 1.7; $p$ = 0.018). Likewise, the students from the financially distressed families by the COVID-19 pandemic suffered from more depression (AOR = 1.7; $p$ = 0.023) and anxiety (AOR = 1.57; $p$ = 0.055), respectively, than the students from the economically unaffected family students. Furthermore, all the potential factors that might impact the students' psychological health of university students during the COVID-19 lockdown in Bangladesh were sketched in Figs 1 and 2.

## Discussion

The present study reported the prevalence of anxiety and depression by 53.5% and 54.6%, respectively, among university students, closely resembling the previous study in Bangladesh. A recent study, which was conducted among students, showed that more than half population of the study sustained from depression (69.5%) and anxiety (61%) [30]. Compared to the earlier studies worldwide, our findings exhibited a higher percentage of anxiety and depression than the global students-based data [31–34]. Before the current COVID-19 pandemic, the prevalence of anxiety and depressive symptoms among youth was 11.6% and 12.9%, respectively [9]. The extended workless and unsubstantial economy might be considered the most common remarkable stressors, resulting in a higher incidence of anxiety and depression in low-income or low-middle-income countries like Bangladesh. A study recommended that ones' maneuver, acquirement, and contentment were closely related to mental well-being, which was greatly impeded by unemployment [35]. In addition, studies revealed that unemployed students without family support limited the individual's self-assurance via affecting the mental health [36]. Besides, the ambiguity regarding online classes, exams, university reopening, communal distancing, endless lockdown may be responsible for this mental illness among university students [12, 25].

The incidence of anxiety and depression disorder has remarkably fluctuated with gender variations in this study. The male university students by 60% and 57.4% were affected by anxiety and depression, respectively, compared with female university students by 44.6% and 50.8%. Similar findings with our study were observed in Egyptian research that male students suffered more depression than females [37]. Moreover, another study postulated that boys (28%) experienced more frequency of depression compared to girls (23%) [34]. From the earlier study in Pakistan, similar findings were reported regarding anxiety disorder. It was observed that male university students of Pakistan were exhibited elevated levels of anxiety

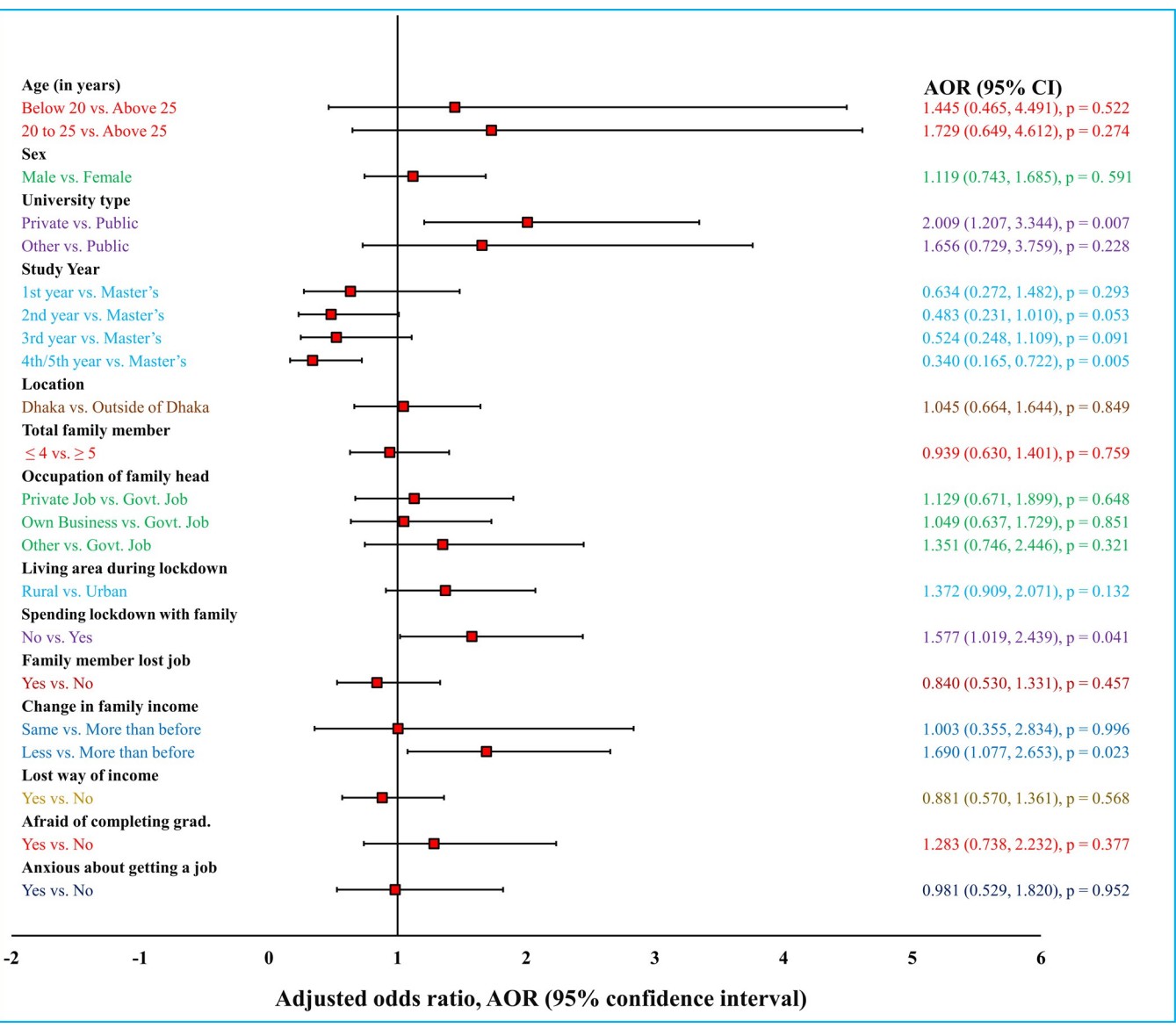

**Fig 1. Forest plot representation for multivariate logistic regression analysis of depression among the Bangladeshi university students amid the second wave of the COVID-19 pandemic.**

than female university students [38]. However, our results are contradicted with several studies in China and Iraq during the pandemic of COVID-19, which revealed that the prevalence of stress, anxiety, and depression was higher in women [39, 40]. However, various studies on mental well-being exhibited no notable gender differences in several countries, including Malaysia [41] and Pakistan [42]. Even though the current study's findings revealed significant anxiety disorder in males, data from a similar past study has shown no significant gender variations [7]. The probable reason behind this finding may be male university students in Bangladesh have undergone more severe oppression regarding graduation, job life, relationships, wedding than their female counterparts.

Private university students revealed a significant incidence of depression and anxiety compared to the public and other universities in Bangladesh. Although, private universities in

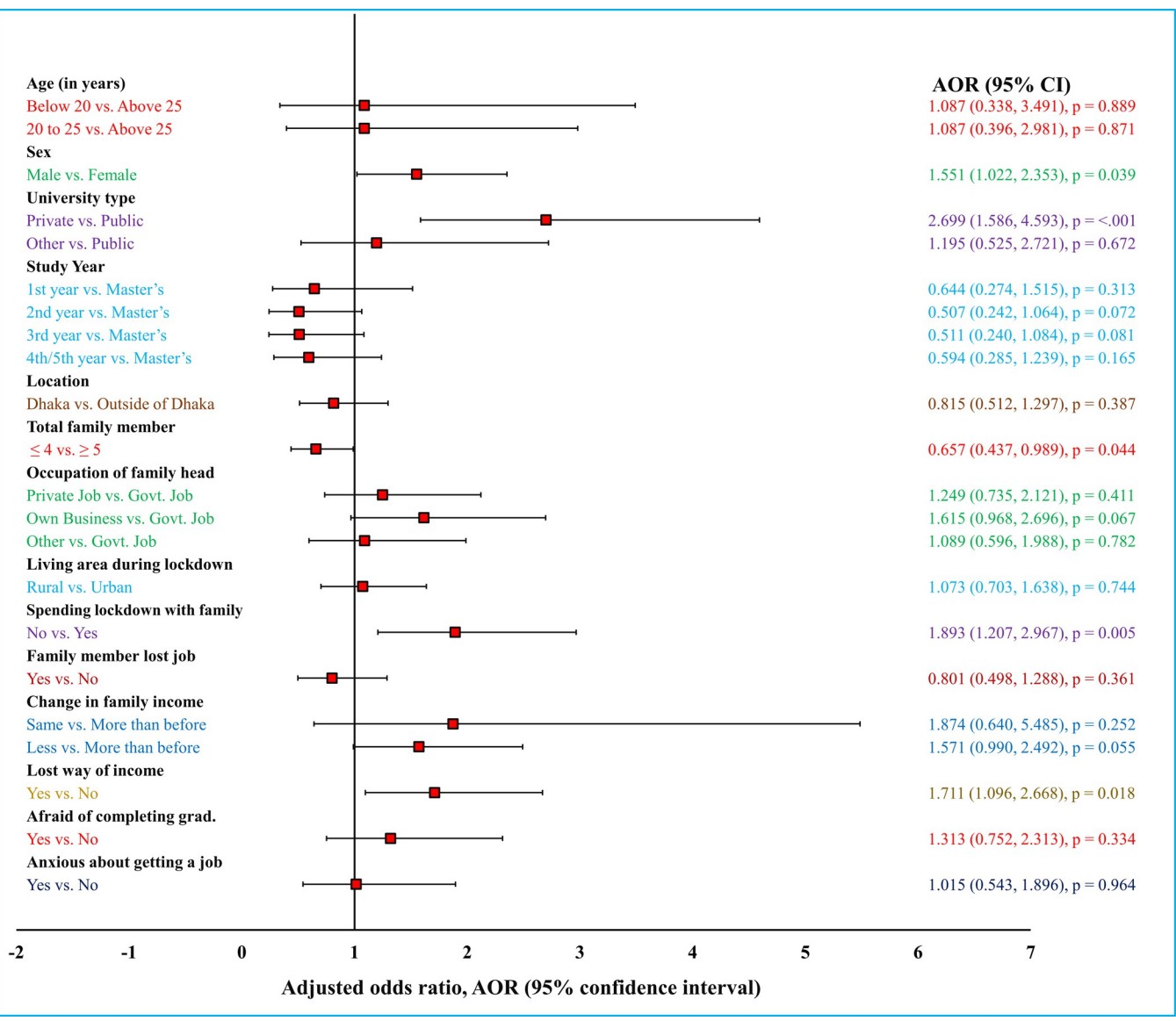

**Fig 2. Forest plot representation for multivariate logistic regression analysis of anxiety among the university students amid the second wave of the COVID-19 outbreak in Bangladesh.**

Bangladesh took the initiative in commencing online classes to surpass the arising problems regarding the extended period of discontinuance of higher education. However, this unprecedented pandemic made more unemployment among the family members, and expenses of private universities became a burden for a family; hence, many students from private universities could not continue their semesters with their classmates. These high educational expenses, apprehension, and falling behind might be attributed to the higher prevalence of anxiety and depression among private university students. Nevertheless, near 80% students could not satisfy regarding the distance learning from the online classes that did not provide adequate knowledge [18, 43], and a remarkable percentage of the rural students could not be able to participate in the online class due to the lacking of digital devices, poor internet connection, and overall economic crisis [18].

The current survey also asserted that pupils of rural areas were more likely to experience more mental health issues, including depressive ailment, compared to urban students. The basis behind this framework is that a longer period of home confinement aggregates the fright of graduation delay, job insecurity, less income, loss of income sources, and eventually financial crisis in rural families than the urban areas [18]. A previous study demonstrated a remarkable finding that living in an urban area was a protective factor from anxiety [44]. Besides, the population in rural areas is moved freely without obeying safety protocols, such as wearing face mask that accelerates the torment of spreading contagion among university students via influencing mental health well-being than the urban populations [45].

Furthermore, the current findings supported that students from fewer family members were significantly associated with anxiety disorder compared to students from higher family members. It may be due to the family support, and more interactions with them could alleviate an individual's mental health sufferings which is possible in families with larger number of members. Previous investigation has shown that vigorous bonding with family members and friends is possible within this pandemic situation; around 60% of populations felt a greater affectionate attitude from their families during this lockdown period compared to ordinary times [46]. Those students who have to live without family during these pandemic conditions and those who have no family are more vulnerable to developing anxiety and depression. Our findings showed significant anxiety and depression ailments among university students associated with how they spend their lockdown period. The students away from families were more mentally distressed than the students who were passing their time with families, and the hypothesis was endorsed by previous evidence [47].

The current results also proposed that the mental health-related problems among university students are related to other socioeconomic variables such as alterations of family income. It exhibited significant responses concerning mental disorders, including anxiety and depression, along with the consequence of the COVID-19 pandemic on individual income and sources of income. A recent study in Bangladesh is accompanying our data by demonstrating a strong relationship with the socioeconomic status of students who belong to a lower-class family suffering from severe depression than pupils coming from middle and higher-class families [18]. The anxiety disorder is more prominent among university students who have undergone unstable family economic conditions due to income sources [23]. Many studies reported that the economic crisis significantly accelerated the increase of common mental health complications [48, 49]. The prevalence of major depressive disorder was found to be increased 8.2% in 2011 from 3.3% in 2008 after the 2009 economic crisis in Greece [50]. Study participants expressing serious financial hardship were most at risk of developing the symptoms of major depressive disorder [48, 50].

In this current report, some socio-demographic variable factors, including age, study year, current location, occupation of leading family members, showed no significant connections with mental health issues, including anxiety and depression. From a previous study, it has been concluded that age and marital status were not remarkable variable factors for student's mental well-being [6]. Besides, our current data is contrasted with several previous studies that exhibited the anxiety and depression levels were greatly varied with the age of students [51, 52].

## Limitations

The recent study has experienced several limitations. Firstly, the sample size of this study was insufficient to represent the whole scenario of the mental health status of university students in Bangladesh. Secondly, study populations were selected and accessed from social media; hence, those who spontaneously took part in research may have satisfactory mental well-being.

Therefore, the prevalence of anxiety and depression was comparatively low. Thirdly, the study was conducted based on a self-reporting technique that generating unnecessary biased results because of convivial gracefulness, individual's mood, and extensive feedback from the respondents. Fourthly, the study did not include the participants who had clinical psychological disturbances from the pre-COVID-19 time. Moreover, it is impossible to characterize the underlying causes of the research as the investigation was a cross-sectional survey.

## Implications and future research

The current study has investigated two major psychological disturbances of Bangladeshi university students following the prolonged COVID-19 lockdown. The results obtained in the study might be representative of the university students' current mental health complications of the other countries having equivalent socioeconomic and cultural characteristics. The research findings will help the policymakers, government, educationalists, and university authorities determine the prevalence of these psychological disorders (depression and anxiety) and their triggering factors. The current statistics will also help the administrators to resolve the severe mental health complications of the university students by taking several effective measures, like implementing professional counseling, economic and job support, and ensuring smooth educational activities. However, several studies need to be conducted to compare the prevalence of the mental health complications of university students with other age groups of people or professionals. Besides, the new normal trends like online classes and online exam systems are equally efficacious for all learners with distinct socioeconomic conditions.

## Conclusion

Despite having some drawbacks, this questionnaire-based survey utilized two standardized and accepted scales to evaluate Bangladeshi university students' mental conditions and reported that more than half of the total respondents are still undergoing MDD and GAD prevalence due to the ongoing COVID-19 catastrophe. In addition to academic and social puzzles, unmanageable economic paucity and professional uncertainty are thrusting upward growth of MDD and GAD among university students. In this circumstance, government, university authorities, and parents must adopt some adamant measures for financial solvency and augmentation of job alliance and ensure an amenable educational environment to alleviate the current psychological mess.

## Supporting information

**S1 Table. The PHQ-9 (PHQ = Patient Health Questioner) and GAD-7 (GAD = Generalized Anxiety Disorder) items to investigate the major depressive disorder and generalized anxiety disorder among the Bangladeshi university students following the extended COVID-19 lockdown.**
(DOCX)

**S1 Data. Raw data file for investigating major depressive disorder and generalized anxiety disorder among the university students following the extended COVID-19 lockdown in Bangladesh.**
(XLSX)

## Author Contributions

**Conceptualization:** Md. Jamal Hossain, Foyez Ahmmed.

**Data curation:** Md. Jamal Hossain, Foyez Ahmmed, Asaduzzaman Hridoy, Farhana Alam Ripa, Talha Bin Emran, Md. Rabiul Islam, Saikat Mitra, Morshed Alam.

**Formal analysis:** Md. Jamal Hossain, Foyez Ahmmed.

**Funding acquisition:** Md. Jamal Hossain, Asaduzzaman Hridoy, Farhana Alam Ripa, Talha Bin Emran.

**Investigation:** Md. Jamal Hossain, Foyez Ahmmed, Labony Khandokar, S. M. Abdur Rahman, Asaduzzaman Hridoy, Farhana Alam Ripa, Talha Bin Emran, Md. Rabiul Islam, Saikat Mitra.

**Methodology:** Md. Jamal Hossain, Foyez Ahmmed, S. M. Abdur Rahman, Asaduzzaman Hridoy, Farhana Alam Ripa, Talha Bin Emran, Md. Rabiul Islam, Saikat Mitra.

**Project administration:** Md. Jamal Hossain, Foyez Ahmmed, S. M. Abdur Rahman, Asaduzzaman Hridoy, Farhana Alam Ripa, Talha Bin Emran.

**Resources:** Md. Jamal Hossain, Foyez Ahmmed, Labony Khandokar, S. M. Abdur Rahman, Asaduzzaman Hridoy, Farhana Alam Ripa, Talha Bin Emran, Md. Rabiul Islam, Saikat Mitra, Morshed Alam.

**Software:** Md. Jamal Hossain, Foyez Ahmmed, Labony Khandokar, S. M. Abdur Rahman, Asaduzzaman Hridoy, Farhana Alam Ripa, Talha Bin Emran, Md. Rabiul Islam, Saikat Mitra, Morshed Alam.

**Supervision:** Md. Jamal Hossain, Foyez Ahmmed, S. M. Abdur Rahman.

**Validation:** Md. Jamal Hossain, Foyez Ahmmed, Labony Khandokar, S. M. Abdur Rahman, Asaduzzaman Hridoy, Farhana Alam Ripa, Talha Bin Emran, Md. Rabiul Islam, Saikat Mitra, Morshed Alam.

**Visualization:** Md. Jamal Hossain, Foyez Ahmmed, Labony Khandokar, S. M. Abdur Rahman, Asaduzzaman Hridoy, Farhana Alam Ripa, Talha Bin Emran, Md. Rabiul Islam, Saikat Mitra, Morshed Alam.

**Writing – original draft:** Md. Jamal Hossain, Labony Khandokar.

**Writing – review & editing:** Md. Jamal Hossain, S. M. Abdur Rahman, Talha Bin Emran, Md. Rabiul Islam.

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
