## [Decision Letter · Decision Letter 0]

13 Dec 2021

PGPH-D-21-00846

Effects of extended closure of university on students’ psychological health in Bangladesh: Results from a cross-sectional pilot study

Dear Dr. Jamal Hossain,

Thank you for submitting your manuscript to PLOS Global Public Health. After careful consideration, we feel that it has merit but does not fully meet PLOS Global Public Health’s publication criteria as it currently stands. Therefore, we invite you to submit a revised version of the manuscript that addresses the points raised during the review process. Please address the reviewers comments, in particular comments about revising the methods and locate the article more deeply in the literature. I also recommend conducting a close reading and revision of the article for grammar, language, and clarity. Please also revise table 1 and only highlight pertinent variables such as the GAD scores etc. and not the questions that fed into GAD scores (these could potentially be shared in an appendix instead) similarly for PHQ.

We look forward to receiving your revised manuscript.

Kind regards,

Anushka Ataullahjan

Guest Editor

Journal Requirements:

1. Please update the completed 'Competing Interests' statement. If you have no competing interests to declare, please state “The authors have declared that no competing interests exist”.

2. In the online submission form, you indicated that "Further details on the raw or extracted datasets in this research are available without restriction from corresponding author.". All PLOS journals now require all data underlying the findings described in their manuscript to be freely available to other researchers, either 1. In a public repository, 2. Within the manuscript itself, or 3. Uploaded as supplementary information.

3. Please provide separate figure files in .tif or .eps format only and remove any figures embedded in your manuscript file. Please ensure that all files are under our size limit of 20MB.  

Once you've converted your files to .tif or .eps, please also make sure that your figures meet our format requirements.

Additional Editor Comments (if provided):

Thank you for sharing this article, I agree with the reviewer comments about revising the methods and more heavily referencing the article. I also recommend conducting a close reading and revision of the article for grammar, language, and clarity. Please also revise table 1 and only highlight pertinent variables such as the GAD scores etc. and not the questions that fed into GAD scores (these could potentially be shared in an appendix instead) similarly for PHQ.

Reviewers' comments:

Reviewer's Responses to Questions

**Comments to the Author**

1. Does this manuscript meet PLOS Global Public Health’s publication criteria? Is the manuscript technically sound, and do the data support the conclusions? The manuscript must describe methodologically and ethically rigorous research with conclusions that are appropriately drawn based on the data presented.

Reviewer #1: Yes

Reviewer #2: Partly

2. Has the statistical analysis been performed appropriately and rigorously?

Reviewer #1: Yes

Reviewer #2: No

3. Have the authors made all data underlying the findings in their manuscript fully available (please refer to the Data Availability Statement at the start of the manuscript PDF file)?

Reviewer #1: Yes

Reviewer #2: Yes

4. Is the manuscript presented in an intelligible fashion and written in standard English?

Reviewer #1: No

Reviewer #2: No

5. Review Comments to the Author

Reviewer #1: Overall comment: This is an important piece of work and while the study appears to be sound, the language is unclear, making it difficult to follow. I advise the authors work with a writing coach or copyeditor to improve the flow and readability of the text.

Abstract: Is well written, however the methods indicate univariate, bivariate, and multivariate analyses were conducted, but no specific mention in the results of the abstract.

Introduction: Should be more focused on the Bangladeshi situation. Recommendation to reduce, especially text around global trends.

Methods: Did you collect data on previous mental health condition before COVID-19? Or if students had COVID-19 or exposure to COVID-19? The literature suggests these are important covariates/confounders.

Results: Are well described.

Discussion: Is quite lengthy and would benefit from a read through to reduce. Limitations were well addressed. Should include a point on previous mental health conditions and exposure/COVID-19 + if these data weren't collected. This could alter findings. Should also put this study in context. Are there any recommendations for future research, programs or policy based on your findings?

Referencing: Some inconsistencies and sentences that should have a citation.

Reviewer #2: Please make major corrections. I have made comments inside the manuscript. please check. Study design, rationale, objectives, tools, statistical analysis are not well- described. English is poor. Tools are used in clinical set-up.

Clear all the above issues.

6. PLOS authors have the option to publish the peer review history of their article (what does this mean?). If published, this will include your full peer review and any attached files.

**Do you want your identity to be public for this peer review?** For information about this choice, including consent withdrawal, please see our Privacy Policy.

Reviewer #1: No

Reviewer #2: **Yes: **Prof. (Dr.) Farzana Saleh

---

## [Decision Letter · Decision Letter 1]

8 Feb 2022

PGPH-D-21-00846R1

Effects of extended university closure on students’ psychological health in Bangladesh: Results from a web-based cross-sectional study

Dear Dr. Hossain,

Thank you for submitting your manuscript to PLOS Global Public Health. After careful consideration, we feel that it has merit but does not fully meet PLOS Global Public Health’s publication criteria as it currently stands. Therefore, we invite you to submit a revised version of the manuscript that addresses the points raised during the review process.

We look forward to receiving your revised manuscript.

Kind regards,

Anushka Ataullahjan

Guest Editor

Additional Editor Comments (if provided):

Reviewers' comments:

Reviewer's Responses to Questions

**Comments to the Author**

1. If the authors have adequately addressed your comments raised in a previous round of review and you feel that this manuscript is now acceptable for publication, you may indicate that here to bypass the “Comments to the Author” section, enter your conflict of interest statement in the “Confidential to Editor” section, and submit your "Accept" recommendation.

Reviewer #2: All comments have been addressed

Reviewer #3: All comments have been addressed

Reviewer #4: All comments have been addressed

2. Does this manuscript meet PLOS Global Public Health’s publication criteria? Is the manuscript technically sound, and do the data support the conclusions? The manuscript must describe methodologically and ethically rigorous research with conclusions that are appropriately drawn based on the data presented.

Reviewer #2: Yes

Reviewer #3: Yes

Reviewer #4: No

3. Has the statistical analysis been performed appropriately and rigorously?

Reviewer #2: Yes

Reviewer #3: Yes

Reviewer #4: Yes

4. Have the authors made all data underlying the findings in their manuscript fully available (please refer to the Data Availability Statement at the start of the manuscript PDF file)?

Reviewer #2: Yes

Reviewer #3: Yes

Reviewer #4: No

5. Is the manuscript presented in an intelligible fashion and written in standard English?

Reviewer #2: Yes

Reviewer #3: Yes

Reviewer #4: No

6. Review Comments to the Author

Reviewer #2: (No Response)

Reviewer #3: Overall, this is a good piece of work. The authors have methodically examined the topic. The manuscript appears sound after revision.

Reviewer #4: (No Response)

7. PLOS authors have the option to publish the peer review history of their article (what does this mean?). If published, this will include your full peer review and any attached files.

**Do you want your identity to be public for this peer review?** For information about this choice, including consent withdrawal, please see our Privacy Policy.

Reviewer #2: **Yes: **Farzana Saleh

Reviewer #3: No

Reviewer #4: No

---

## [Decision Letter · Decision Letter 2]

9 Mar 2022

Status of psychological health of students following the extended university closure in Bangladesh: Results from a web-based cross-sectional study

PGPH-D-21-00846R2

Dear Dr. Hossain,

We are pleased to inform you that your manuscript 'Status of psychological health of students following the extended university closure in Bangladesh: Results from a web-based cross-sectional study' has been provisionally accepted for publication in PLOS Global Public Health.

Before your manuscript can be formally accepted you will need to complete some formatting changes, which you will receive in a follow up email. A member of our team will be in touch with a set of requests. There are also several minor comments that need to be resolved.

Best regards,

Anushka Ataullahjan

Guest Editor

Reviewer Comments (if any, and for reference):

Reviewer's Responses to Questions

**Comments to the Author**

1. If the authors have adequately addressed your comments raised in a previous round of review and you feel that this manuscript is now acceptable for publication, you may indicate that here to bypass the “Comments to the Author” section, enter your conflict of interest statement in the “Confidential to Editor” section, and submit your "Accept" recommendation.

Reviewer #3: All comments have been addressed

Reviewer #4: All comments have been addressed

2. Does this manuscript meet PLOS Global Public Health’s publication criteria? Is the manuscript technically sound, and do the data support the conclusions? The manuscript must describe methodologically and ethically rigorous research with conclusions that are appropriately drawn based on the data presented.

Reviewer #3: Yes

Reviewer #4: Yes

3. Has the statistical analysis been performed appropriately and rigorously?

Reviewer #3: Yes

Reviewer #4: Yes

4. Have the authors made all data underlying the findings in their manuscript fully available (please refer to the Data Availability Statement at the start of the manuscript PDF file)?

Reviewer #3: Yes

Reviewer #4: Yes

5. Is the manuscript presented in an intelligible fashion and written in standard English?

Reviewer #3: Yes

Reviewer #4: Yes

6. Review Comments to the Author

Reviewer #3: The manuscript appears good after revision.

Reviewer #4: (No Response)

7. PLOS authors have the option to publish the peer review history of their article (what does this mean?). If published, this will include your full peer review and any attached files.

**Do you want your identity to be public for this peer review?** For information about this choice, including consent withdrawal, please see our Privacy Policy.

Reviewer #3: No

Reviewer #4: No
